# Biomineralization of Polyelectrolyte-Functionalized Electrospun Fibers: Optimization and In Vitro Validation for Bone Applications

**DOI:** 10.3390/biomimetics9040253

**Published:** 2024-04-22

**Authors:** Ahmed Salama, Emad Tolba, Ahmed K. Saleh, Iriczalli Cruz-Maya, Marco A. Alvarez-Perez, Vincenzo Guarino

**Affiliations:** 1Cellulose and Paper Department, National Research Centre, 33 El Bohouth St., Dokki, Giza 12622, Egypt; asrk_saleh@yahoo.com; 2Polymers and Pigments Department, National Research Centre, 33 El-Buhouth St., Dokki, Giza 12622, Egypt; 3Institute of Polymers, Composite and Biomaterials, National Research Council of Italy, Mostra d’Oltremare, V.le J.F. Kennedy 54, 80125 Naples, Italy; iriczalli.cruzmaya@ipcb.cnr.it; 4Tissue Bioengineering Laboratory, DEPeI, School of Dentistry, Universidad Nacional Autonoma de Mexico (UNAM), Circuito Exterior s/n C.P., Mexico City 04510, Mexico; marcoalv@unam.mx

**Keywords:** cationic cellulose, electrospun fibers, antimicrobial properties, in vitro differentiation, osteogenic response

## Abstract

In recent years, polyelectrolytes have been successfully used as an alternative to non-collagenous proteins to promote interfibrillar biomineralization, to reproduce the spatial intercalation of mineral phases among collagen fibrils, and to design bioinspired scaffolds for hard tissue regeneration. Herein, hybrid nanofibers were fabricated via electrospinning, by using a mixture of Poly ɛ-caprolactone (PCL) and cationic cellulose derivatives, i.e., cellulose-bearing imidazolium tosylate (CIMD). The obtained fibers were self-assembled with Sodium Alginate (SA) by polyelectrolyte interactions with CIMD onto the fiber surface and, then, treated with simulated body fluid (SBF) to promote the precipitation of calcium phosphate (CaP) deposits. FTIR analysis confirmed the presence of SA and CaP, while SEM equipped with EDX analysis mapped the calcium phosphate constituent elements, estimating an average Ca/P ratio of about 1.33—falling in the range of biological apatites. Moreover, in vitro studies have confirmed the good response of mesenchymal cells (hMSCs) on biomineralized samples, since day 3, with a significant improvement in the presence of SA, due to the interaction of SA with CaP deposits. More interestingly, after a decay of metabolic activity on day 7, a relevant increase in cell proliferation can be recognized, in agreement with the beginning of the differentiation phase, confirmed by ALP results. Antibacterial tests performed by using different bacteria populations confirmed that nanofibers with an SA-CIMD complex show an optimal inhibitory response against *S. mutans*, *S. aureus*, and *E. coli*, with no significant decay due to the effect of CaP, in comparison with non-biomineralized controls. All these data suggest a promising use of these biomineralized fibers as bioinspired membranes with efficient antimicrobial and osteoconductive cues suitable to support bone healing/regeneration.

## 1. Introduction

In the wide range of biomimetic strategies promoting bone repair, the use of calcium phosphate materials currently represents the gold standard for replacing injured bone because they resemble bone mineral crystals [1,2]. As nature teaches, natural bones are composed of hierarchical nanocomposite structures resulting from a strict packing of hydroxyapatite nanocrystals into collagen fibrous matrix, forming an extracellular matrix [3]. Today, the design of new grafting for bone repair/regeneration is inspired by materials and technologies that reproduce the structural composition of natural bone tissue to trigger, to some extent, the biomineralization processes that concur to form hierarchical bone-like structures [4]. In detail, collagen fibrils serve as nucleation sites for forming hydroxyapatite crystals in the bottom-up biomineralization process. In order to attain a high degree of structural homogeneity, numerous studies have been conducted to create biomimetic nanocomposite materials for medical applications. An interesting approach involves using in situ reactions in organic solutions to form inorganic compounds from their precursor solutions in situ [5,6,7]. However, relevant problems still concern the control of spatial confinement and the agglomeration tendency of inorganic phases into the matrix, especially in the case of submicrometric structures based on micro- and/or nanofibers assemblies [8]. In this view, one of the most adaptable techniques is depositing inorganic phases to form biomimetic coatings that can efficiently improve the bioactivity of orthopedic implant surfaces [9].

Several studies have suggested the use of bioactive treatments to make biologically active the surface of polymeric substrates. This is crucial to create optimal microenvironmental conditions for cells, by promoting a more efficient interfacial bonding with the surrounding tissues and limiting the uncontrolled formation of fibrous tissue at the interface, which may be relevant in terms of failure or implant instability. In particular, a large amount of experimental evidence has confirmed that the development of a bone-like layer onto polymeric surfaces using Simulated body fluid (SBF)—a synthetic medium containing an ion concentration near to that of human blood plasma—concurs to expose biochemical signals able to significantly improve the bioactivity of polymeric matrices generally used as scaffolds for bone regeneration [10,11,12,13,14].

For instance, Coelho et al. used SBF solutions of different ionic strengths to coat the surface of titanium and related alloys [14]. The results demonstrate the deposition of a stable apatite-like layer with a Ca/P ratio of 1.65, which is close to the value of biological apatite. In addition, the surface of polymeric materials was also functionalized with a biomimetic calcium phosphate layer to improve the osteogenic features of polymeric scaffolds. Yang et al. [15] reported the ability of surface-modified poly L-lactic acid (PLLA) with bioactive polydopamine (PDA) in a basic condition to allow the deposition of a bone-like calcium phosphate via improving surface wettability. Liao and Ho examine the potential of chitosan scaffold to promote the deposition of calcium phosphate from SBF. In this study, two crosslinking agents, glutaraldehyde (GA) and sodium tripolyphosphate (TPP), were used [16]. The results indicate that the fabricated chitosan scaffolds crosslinked with TPP provide appropriate surface properties for the growth of calcium phosphate and the spreading, growth, and alkaline phosphate (ALP) expression of osteoblast-like cells. 

In recent years, cellulose-based materials have attracted considerable attention as the highest candidate feedstock for natural polymeric material production [17]. Hence, significant progress has been achieved in developing novel biopolymers based on cellulose and its derivatives [9,18]. The use of different cellulosic forms—namely, fibers or nanocrystals—as filler in nanocomposite materials fabrication presents an alternative route to produce sustainable polymer-based composites and functional biocompatible products. Cellulose has a one-dimensional anisotropic crystalline nanostructure with spectacular mechanical strength, biocompatibility, and renewability. Moreover, it can be modified through chemical functionalization to produce anionic or cationic derivatives, which makes cellulose a promising bio-template for controlling the mineralization of inorganic particles into hierarchical nanostructured products. Several trials have been conducted for preparing cellulose calcium phosphate hybrids [19,20]. For example, TEMPO-oxidized cellulose nanofibers were grafted with soy protein hydrolysate via amidation of carboxylic groups and evaluated as bioactive biopolymers through calcium phosphate (CaP) precipitation in SBF [19]. The results exhibited the precipitation of highly crystalline calcium phosphate with a similar composition to those of native apatites of bone (Ca/P ratio equal to 1.63).

Moreover, the healing of bone defects is closely associated with the microstructure features and biological performance of tissue scaffolding materials to provide an optimal bone regeneration microenvironment. However, the fabrication of implants with biomimetic signals able to reproduce the hard tissue microenvironment and support their ingrowth remains a partially unsolved challenge. Herein, PCL/polyelectrolyte hybrid membranes were fabricated via electrospinning and self-assembly strategies, by incorporating a new cationic cellulose derivative and sodium alginate into PCL nanofibers.

PCL electrospun nanofibers were widely used in bone tissue engineering due to their optimal mechanical properties (i.e, toughness, mechanical stiffness) that tend to be stable under physiological conditions for several months, as a consequence of their slow degradation in biological fluids [21,22]. Their combination with natural polysaccharides such as cellulose and sodium alginate improves fiber biocompatibility, contributing to the creation of a substrate highly friendly for cells and able to facilitate the transport of molecules, drugs, and tissue morphogens [23]. A recent work by our group has just demonstrated that CIMD in combination with SA shows interesting antibacterial properties and a good in vitro response for wound healing applications [24]. In this context, it was proposed to functionalize composite fibers with a bone-like calcium phosphate layer via biomimetic treatment in order to overcome the intrinsically limited activity of these polyelectrolytes to support osteogenesis [25]. In this work, hybrid nanofibers were deeply investigated in terms of chemical structure and composition, morphology, in vitro biocompatibility, and antimicrobial properties to validate their use for bone healing and regeneration.

## 2. Results and Discussion

### 2.1. Synthesis and Characterization of Cellulose Derivate 

Tosyl cellulose and 1-methylimidazole were reacted to form a cellulose derivative with cationic soluble in water—namely, imidazolium tosylate, as reported in our previous study [24]. Various analytical methods, including elemental analysis and ^13^C NMR, proved cationic cellulose formation.

All the samples—i.e., before and after mineralization—were investigated by FT–IR spectroscopy (Figure 1). PCL showed a main adsorption peak at 2900 cm^−1^, due to -CH_2_ stretching vibrations, and 1722 cm^−1^, related to ν(C=O). Moreover, the bending modes (δ) of -CH_2_ groups were detected at 1472 cm^−1^, and ν(C-O) were detected at 1172 cm^−1^ [26]. In the presence of CIMD, characteristic bands at 1161 cm^−1^ (υs SO_2_), 1361 cm^−1^ (υas SO_2_), and 1539 cm^−1^ (ν C=C aromatic) were recognized, associated with the tosyl groups. In the presence of SA, the characteristic adsorption peaks of alginates were also detected: a broad band related to the hydroxyl groups’ stretching vibrations (ν), close to 3450 cm^−1^, and other bands at 1599 cm^−1^ and 1417 cm^−1^, respectively, were related to asymmetric (ν_a_) and symmetric (ν_s_) stretching of the carboxylic groups. After immersing in the SBF solution, the characteristic peaks of calcium phosphate groups related to vibration modes of the PO_4_^3−^ groups started to appear, respectively, at 1066 cm^−1^ (P–O ν3 mode), 579 cm^−1^ (P–O ν4 mode), and 933 cm^−1^ (P–O ν1 mode).

### 2.2. Biomimetic Mineralization of the Functionalized PCL Fiber Mats

The technique of biomimetic mineralization has been expanded to create calcium phosphate coatings on biodegradable polymer substrates that resemble biological bone [27]. This study aimed to prepare a biomimetic bone composite material by examining the mineralization of neat PCL electrospun fibers compared to the PCL modified with CIMD and CIMD/SA ones. Figure 2 shows SEM micrographs of all the samples after biomimetic treatment. The PCL/CIMD/SA sample’s surface was entirely covered by CaP precipitates with characteristic nanometric size. Conversely, on the PCL and PCL/CIMD samples, SEM images enabled calcium phosphate precipitates to be recognized at only a few points, with heterogeneous sizes and morphologies, due to the use of a non-invasive mineralization treatment.

According to the previous evidence, the presence of chemical elements such as calcium (Ca) and phosphorus (P) related to the presence of calcium phosphates was verified by energy-dispersive X-ray spectroscopy (EDX). Results for PCL/CIMD/SA and PCL/CIMD/SA/CP are summarized in the elemental map and spectra reported in Figure 3. Only in the case of PCL/CIMD/SA/CP were calcium and phosphorus elements recognized with an average Ca/P ratio, falling into the range of biological apatite octacalcium phosphate—about 1.33—which is a precursor for biological apatite formed during biomineralization [28].

### 2.3. In Vitro Cell Results

Cytocompatibility tests were performed by using hMSCs. As reported in previous work, good cell adhesion and proliferation were recognized in PCL nanofibers modified with CIMD and CIMD/SA, especially in the presence of SA [24]. Herein, the in vitro effect of mineralized phases onto the fibers via additional SBF treatment was investigated. Thus, PCL/CP, PCL/CIMD/CP, and PCL/CIMD/SA/CP samples were considered for in vitro tests. Cell adhesion was evaluated after 4 and 24 h of seeding. Results were reported as % adhesion of cells, calculated with respect to that in tissue culture plate (TCP)—i.e., % adhesion equal to 100% (Figure 4A). After 4 h, no significant differences were identified among the groups. However, after 24 h, there was an increase in cell adhesion on PCL/CIMD/CP and PCL/CIMD/SA/CP nanofibers, reporting a significant difference with respect to PCL/CP fibers; this is also confirmed by SEM images showing adhered cells onto the fibers after 24 h (Figure 4B–D). Moreover, in contact with PCL/CP fibers, cells tend to acquire a rounded shape, while cells seeded onto PCL/CIMD/CP and PCL/CIMD/SA/CP tend to spread along the fibers. This is more evident in the presence of SA, conferring a hydrophilic signal to the fiber surface. In this context, the further presence of calcium phosphates promotes the formation of focal adhesion points, according to previous studies [29], thus corroborating cell attachment mechanisms.

hMSCs proliferation was assessed after 1, 3, and 7, 14 days in the case of PCL/CP, PCL/CIMD/CP, and PCL/CIMD/SA/CP fibers (Figure 5). For this purpose, a colorimetric assay was used to assess cell viability and proliferation as a function of metabolic activity. Results showed an increase in cell viability after just 3 days in all cases, with more remarkable effects in the case of PCL/CIMD/SA/CP due to the hydrophilic contribution of SA, as reported in previous works [24].

On day 7, the metabolic activity was reduced in all the cases, especially in the case of PCL/CIMD/CP and PCL/CIMD/SA/CP. This may be attributable to an initial stage of differentiation triggered by the presence of CaP [30], which induces a metabolic shift from proliferation to differentiation, thus limiting the proliferation [31,32]. Then, cell proliferation increased significantly after 7 days of culture, in agreement with previous studies based on hybrid/composite substrates, including hydroxyapatite-rich phases [33].

In this case, in vitro response is also supported by the porosity of the fibrous network in terms of adhesion and cell proliferation. CaP does not affect the in vitro response of hMSCs but plays a significant role in addressing the osteogenic response for bone. Furthermore, the ability of electrospun nanofibers to mimic the fibrillary organization of the native ECM is well known [34]. In this context, the addition of inorganic solid signals can concur to reproduce the pattern of biochemical signals typically found in the mineralized ECM, which is able to trigger osteogenic differentiation [35,36]. In this view, hMSC differentiation was evaluated after 3 and 7 days using an early marker, i.e., alkaline phosphatase (ALP) (Figure 6). After 3 days, an increase in ALP was recognized in the case of CaP-coated samples with respect to the negative controls (TCP and PCL). However, some differences can be recognized as a function of the fiber composition—e.g., SA and/or CMID, thus confirming the active role of polyelectrolyte functionalization in influencing cell differentiation, over the presence of CaP [37,38]. After 7 days of culture, an increase in ALP activity was recognized in all groups. In particular, the presence of SA concurs to promote a more remarkable increase in ALP activity with respect to only CIMD-treated ones. This is due to the contribution of hydroxyl groups able to support the formation of nucleation sites for the precipitation of calcium phosphates [39,40,41] and improve the interfacial interaction with calcium phosphate through hydrogen bonds [42].

### 2.4. Antimicrobial Evaluation of Composite Films

The disc diffusion method was used for the antimicrobial performance of PCL/CP, PCL/CIMD/CP, and PCL/CIMD/SA/CP composite films against four model microbes. The observed inhibition zone was reported to determine the release potency of the active agent of CIMD, hence, the composite films’ antimicrobial activity. The response of PCL/CP, PCL/CIMD/CP, and PCL/CIMD/SA/CP composite films was investigated on pathogenic strains with the use of the disc diffusion method (Figure 7).

The quantitative results are represented in Table 1. Results demonstrate the antimicrobial activity of PCL/CIMD/CP and PCL/CIMD/SA/CP composite films against all employed pathogens and exhibit higher toxicity against Gram-positive *S. mutans* and *S. aureus* followed by toxicity against Gram-negative *E. coli* and lower toxicity toward *C. albicans*. From the obtained data, we can observe that the composite films containing CP exhibit no significance for enhancing antimicrobial tests; this observation indicates that the activity of composite films is related to the presence of CIMD in the composite films, in agreement with experimental evidence reported in previous works [43,44]. According to the reported results, the activity of PCL as an antimicrobial agent was not relevant as reported in previous studies [45,46]. This can be explained by the effect of methyl and hydroxyethyl groups and the length of the alkyl chain, containing more than ten carbon atoms [47,48,49], which influenced the mechanism of action of CIMD against pathogenic microbes.

## 3. Conclusions

The impact of PCL fiber surface functionalization on calcium phosphate deposition and the biocompatibility characteristics of calcium phosphate-coated fiber mats was studied. These results confirmed a mutual role of biomineralization in combination with SA treatment to mimic the environment of cells, modulating the differentiation of hMSCs toward an osteogenic fate. In addition, the obtained results suggest that the composite films containing CIMD provide a better inhibitory response against Gram-negative and Gram-negative bacteria, which confirms that the composite films may be utilized in biomedical applications.

## 4. Materials and Methods

### 4.1. Materials

Microcrystalline cellulose, p-toluenesulfonyl chloride, sodium alginate (SA), 1-methylimidazole, and Polycaprolactone (Mw 70,000–90,000) were purchased from Sigma-Aldrich, Milan, italy). Formic acid (85%) and acetic acid (96%) from Edwic, Cairo, Egypt, 1-Butyl-3-methylimidazolium chloride from IoLiTec (Heilbronn, Germany). No purification was required for chemicals used, being of analytical quality.

### 4.2. Electrospun Fiber Preparation

Briefly, 1 g of PCL pellets was dissolved in 10 mL mixture of formic acid/acetic acid solvents (70:30 *v*/*v*) to form a clear solution (10% *w*/*v*) under continuous stirring for 120 min. As for blended fibers, a fixed amount of CIMD was dissolved in PCL solution to obtain 80:20 (*w*/*w*). PCL/CIMD relative ratio. PCL and PCL/CIMD solutions were processed via electrospinning by using a homemade system with vertical configuration composed of a syringe pump (KDS Series 100, KD Scientific, MA, US) connected to a metallic needle (19G) to control the solution flow rate and a power supply with single (positive) polarity (Glassman High Voltage Series—working range 0–20 kV) to control the applied voltage. Process parameters were set as follows: Voltage 17 kV, flow rate 0.5 mL/h, needle tip/ground distance of 150 mm. All the tests were performed at room temperature (about 25 °C) and at a relative humidity of 60 ± 19%.

### 4.3. Functionalization via Polyelectrolytes

PCL/CIMD nanofibers were immersed in anionic solution 0.2% SA (*w*/*v*). The resulting samples were soaked in 100 mL of this solution for 20 min and, then, washed in 100 mL of bidistilled water to eliminate additional SA residues. Lastly, hybrid nanofibers were dried at 45 °C in a vacuum oven for 48 h and stored under vacuum for the next characterizations. 

### 4.4. Biomimetic Mineralization

The fabrication of dual-layered bioactive mats of PCL nanofiber and Self-assembled polyelectrolyte complexes of CIMD and SA were prepared according to our previous report. In order to investigate the potential of the prepared fiber mats to promote the deposition of calcium phosphate crystals, a supersaturated solution of simulated body fluid (SBF) was prepared from different salts, including NaCl, CaCl_2_, NaHCO_3_, and Na_2_HPO_4_. The obtained solution was filtered through Millipore^®^ Sterile Filters (0.22 μm pore size). The neat PCL, PCL/CIMD, and PCL/CIMD/SA samples were immersed in 100 mL of SBF in a capped plastic tube and kept at 37 °C overnight. The SBF solution was refreshed every day. After 3 days, the sample was gently washed with water and then dried in air at room temperature.

### 4.5. Fiber Characterization

Morphological properties were qualitatively investigated via field emission SEM (Jeol JXA 840, Tokyo, Japan). Fourier-transform infrared (FTIR) spectroscopy was performed to evaluate the new function group in the prepared samples with an attenuated total reflectance-FTIR spectroscope/Varian (Agilent, Santa Clara; CA, USA). Moreover, the Z potential of the fiber surface was explored through dynamic light scattering (DLS) (Malvern Zetasizer NanoZS, Worcestershire, UK). The nanofibers were suspended in 2 mL of water and examined with a zeta dip cell.

### 4.6. Biocompatibility Assays

#### 4.6.1. Cell Culture

For in vitro evaluation, passages 4–5 of human mesenchymal stem cells (hMSCs, SCC034) were used. hMSCs were cultured in a 75 cm^2^ cell culture flask in Eagle’s alpha minimum essential medium (α-MEM) supplemented with 10% of fetal bovine serum (FBS, Sigma Aldrich, Milan, Italy), antibiotic solution of streptomycin 100 µg/mL and penicillin 100 U/mL (Sigma Aldrich, Milan, Italy), and 2 mM of L-glutamine (Sigma Aldrich, Milan, Italy) inincubation environment of 37 °C in a humidified atmosphere with 5% CO_2_ and 95% air. Once cells reached about 80% of confluence, they were seeded onto the samples, previously sterilized in ethanol (70% *v*/*v* in water), and placed into a 96-cell culture plate.

#### 4.6.2. In Vitro Assays

hMSCs (1 × 10^4^) were cultured on all the samples and left in standard conditions for cell adhesion assay at 4 and 24 h. After the incubation time, unattached cells were removed by withdrawing the medium, and then samples were washed with PBS. Cell Proliferation Kit II (XTT, Roche Diagnostics Deutschland GmbH, Mannheim, Germany) prepared according to the manufacturer’s was used in fresh cell culture medium after sample incubation for 4 h. After this period, the supernatant was recovered and placed in an ELISA plate reader to measure the absorbance at 450 nm (Wallac Victor 1420, PerkinElmer, Boston, MA, USA). The cell adhesion is presented as a percentage with respect to the tissue culture plate (TCP).

The cell morphology of hMsCs seeded on each group of fibers was examined by SEM (scanning electron microscopy, QuantaFEG 200, FEI, Eindhoven, The Netherlands) after 24 h in cell culture. After that period, the samples were washed and fixed with PBS and 4% formaldehyde, respectively. Then, samples were dehydrated with graded series of ethanol (25–100%) and air-dried.

The viability of cells seeded onto all the samples was evaluated by XTT assay kit after 1, 3, 7, and 14 days. After each culture period, the medium was changed to fresh medium containing XTT working solution as indicated by the manufacturer’s instructions, and then incubated for four hours. Subsequently, the supernatant was collected to measure the absorbance at 450 nm. Results are presented as mean ± standard error deviation (*n* = 3). Analysis of variance (ANOVA) with Tukey’s post hoc for differences between groups. A value of *p* < 0.05 was considered to determine statistically significant differences.

To evaluate the influence of fibers on hMSCs differentiation, a colorimetric assay for alkaline phosphatase (ALP) activity (ab83369 Abcamp, Cambridge, UK) was performed after 3 and 7 days in culture. Briefly, samples were washed with PBS to then add 250 µL of lysis buffer (Lysis Buffer R2, Gibco Life Technologies, New York, NY, USA) and transferred to a 1.5 mL. For the assay, 80 μL of the lysate was added to a 96-well plate with 50 μL of 5 mM of pNPP solution and incubated at 25 °C for 60 min. After the incubation time, 20 μL of stop solution was added to stop the reaction. ALP activity was evaluated by measuring the absorbance of the supernatant at 405 nm. The measurement of the activity in the samples was calculated using a standard curve.

### 4.7. Antimicrobial Evaluation by Disc Diffusion Method

The antimicrobial performance of the PCL/CP, PCL/CIMD/CP, and PCL/CIMD–SA/CP mats was qualitatively evaluated by the disc diffusion method, according to our previous study [36], against four pathogenic microbes obtained from American Type Culture Collection (ATCC), including Gram-positive bacteria, Streptococcus mutant ATCC 25175 (*S. mutans*), Gram-negative bacteria Escherichia coli ATCC 25922 (*E. coli*) and Salmonella typhimurium ATCC 14028 (*S. typhimurium*), and *Candida albicans* ATCC 10231 (*C. albicans*) as a yeast model. For standard inoculum preparation, the pure colony from each model microbes was inoculated in a 250 mL conical flask containing 50 mL of sterilized Mueller Hinton broth composed of (%):0.15 starch, 1.75 acid hydrolysate of casein, and 0.2 beef extract, and incubated at 37 °C, for 24 h under shaking conditions at 200 rpm. After the incubation period, about 100 μL of serially diluted pathogens (106 CFU/mL) equal density to 0.5 McFarland standards were inoculated separately on Petri dishes containing 15 mL of sterilized Mueller–Hinton media (20 g/L agar) along with the tested composite films and incubated at 37 °C for 24 h. The resulting inhibitory action was assessed by measuring the inhibition halo zone including the composite films. The composite films were sterilized through ultraviolet light for 30 min. Obtained data are the result of the average values from triplicate measurements.

## Figures and Tables

**Figure 1 biomimetics-09-00253-f001:**
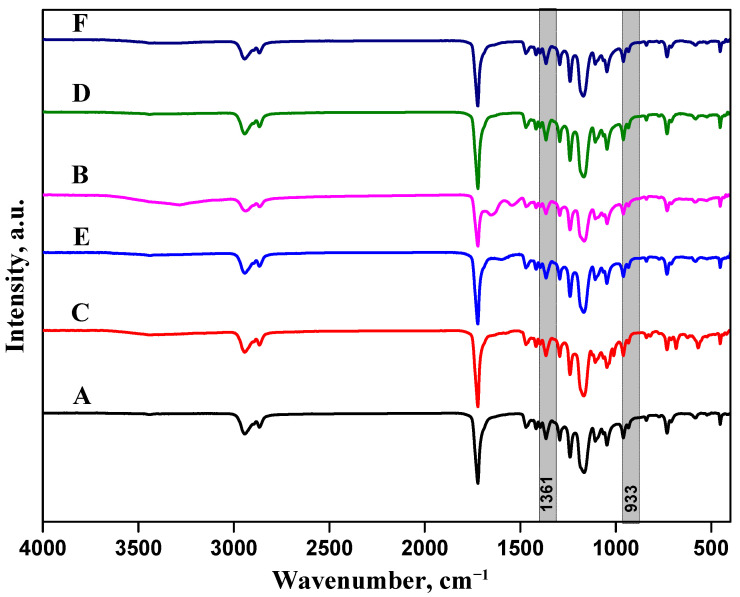
FT-IR spectroscopy of PCL, PCL/CIMD, and PCL/CIMD/SA before (**A**,**C**,**E**) and after calcium phosphate mineralization (**B**,**D**,**F**), respectively. Gray strips highlighted the main characteristic peaks for CIMD and Phosphate groups.

**Figure 2 biomimetics-09-00253-f002:**
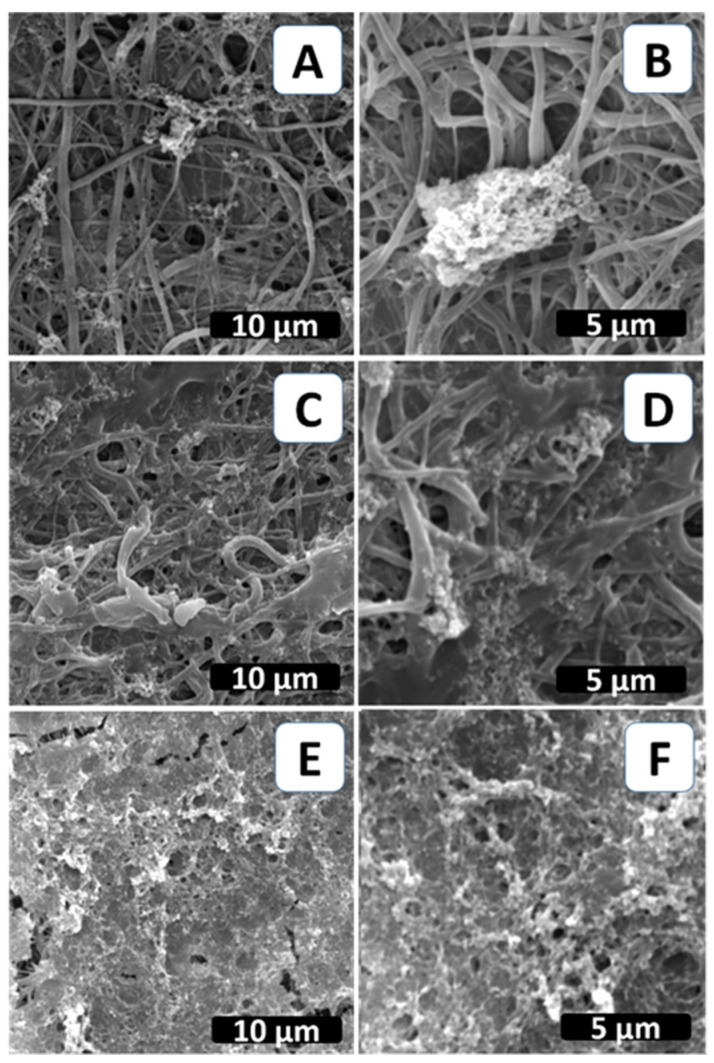
Surface morphology after calcium phosphate mineralization of neat PCL (**A**,**B**), PCL/CIMD (**C**,**D**), and PCL/CIMD/SA (**E**,**F**).

**Figure 3 biomimetics-09-00253-f003:**
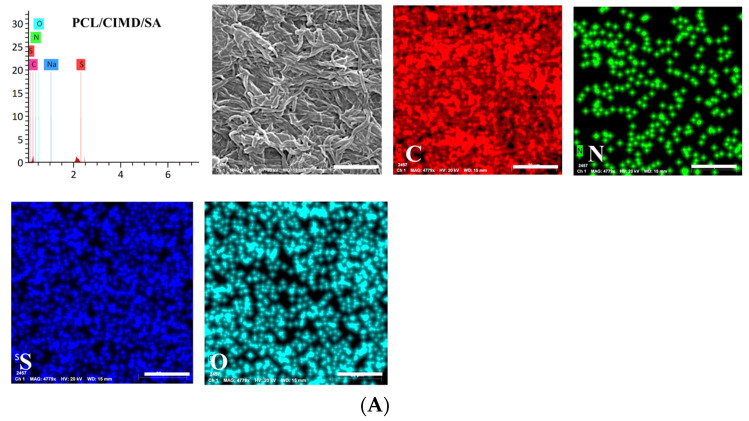
Elemental mapping via EDX for (**A**) PCL/CIMD/SA and (**B**) PCL/CIMD/SA/CP: Carbon (Red), Nitrogel (Green), Sulfur (Blue); Oxygen (Cyan), Phosphorus (Yellow), and Calcium (Orange).

**Figure 4 biomimetics-09-00253-f004:**
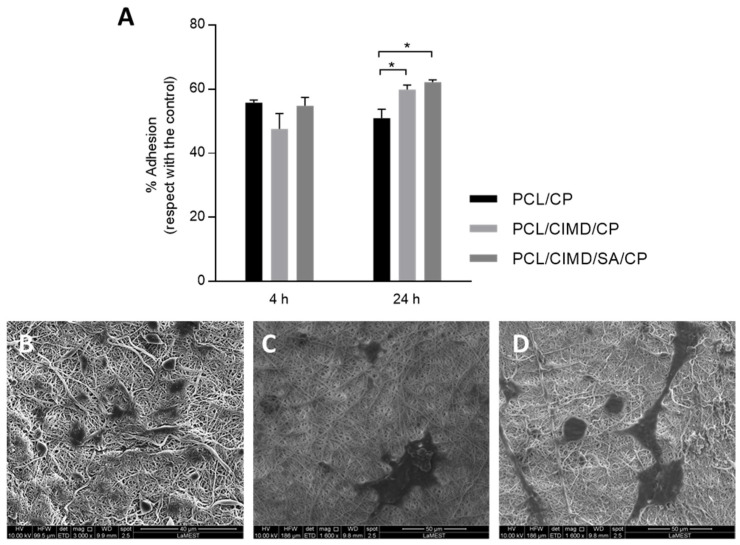
In vitro studies of biocompatibility: (**A**) Adhesion of cells, calculated with respect to tissue culture plate (TCP). Significant differences are represented as * *p* < 0.05. SEM images of hMSCs after 24 h in cell culture with PCL/CP (**B**), PCL/CIMD/CP (**C**), and PCL/CIMD/SA/CP (**D**) nanofibers.

**Figure 5 biomimetics-09-00253-f005:**
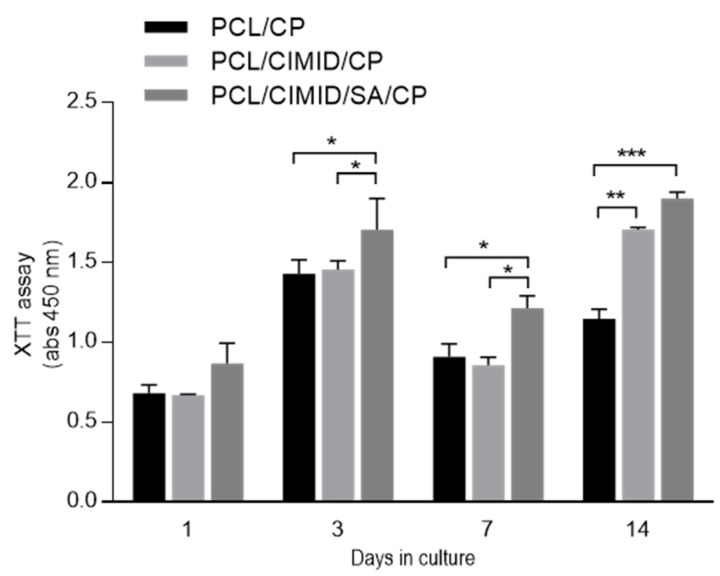
In vitro Viability (XTT tests): hMSC response for 14 days on PCL, PCL/CIMD, and PCL/CIMD/SA after biomineralization. Data were reported as mean ± standard error deviation (* *p* < 0.05, ** *p* < 0.01, *** *p* < 0.001).

**Figure 6 biomimetics-09-00253-f006:**
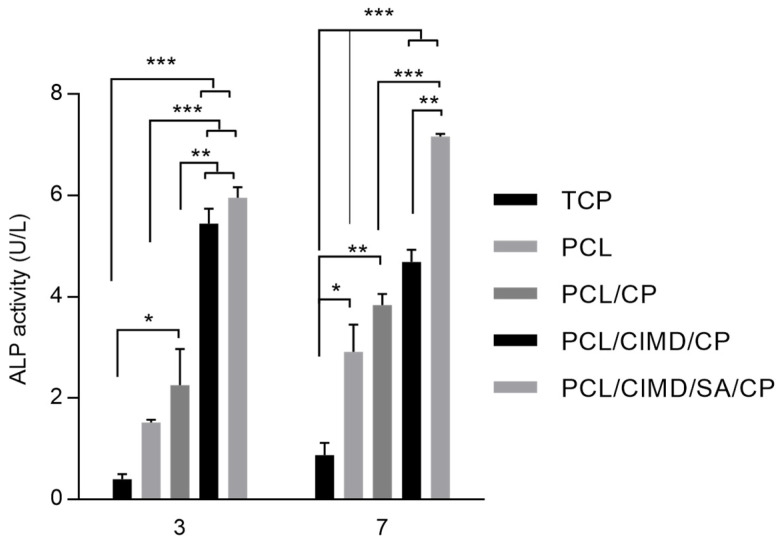
ALP activity of hMSCs after 3 and 7 days of culture onto the tissue culture plate (TCP), PCL, PCL/CP, PCL/CIMD/CP, and PCL/CIMD/SA/CP fibers. (* *p* < 0.05, ** *p* < 0.01, *** *p* < 0.001).

**Figure 7 biomimetics-09-00253-f007:**
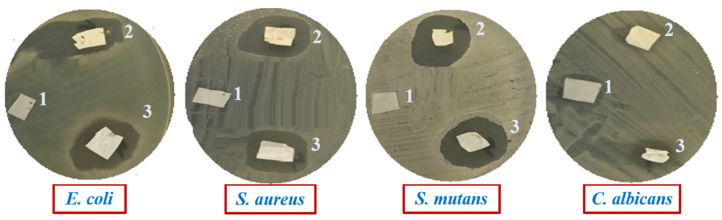
Antimicrobial tests by halo-zone expression, against four different strains. PCL/CP, PCL/CIMD/CP, and PCL/CIMD/SA/CP were coded as 1, 2, and 3, respectively.

**Table 1 biomimetics-09-00253-t001:** Antimicrobial activity of the PCL, PCL/CIMD, and PCL/CIMD/CP composite films against four pathogenic microbes.

Pathogenic Microbes	Diameters of Inhibition Zone (mm)		
	PCL/CP	PCL/CIMD/CP	PCL/CIMD/SA/CP
*E. coli*	0.0 ± 0.0	19 ± 1.17	18 ± 1.07
*S. aureus*	0.0 ± 0.0	20 ± 1.21	19 ± 1.17
*S. mutans*	0.0 ± 0.0	22 ± 1.19	20 ± 1.13
*C. albicans*	0.0 ± 0.0	7 ± 0.06	8 ± 0.08

## Data Availability

The data are contained within the article.

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
