# Peer review of "Biomineralization of Polyelectrolyte-Functionalized Electrospun Fibers: Optimization and In Vitro Validation for Bone Applications"

_biomimetics, 2024, doi:10.3390/biomimetics9040253_

Round 1

Reviewer 1 Report

Comments and Suggestions for Authors

In the manuscript Authors investigated the selected properties of the nanofibers composed of polycaprolactone and cellulose modified with imidazolium tosylate in order to incorporate cationic groups into its chains. Sodium alginate was used as a polymer bearing anionic groups to combine with nanofibers by interactions with CIMD. The product was treated by simulated body fluid (SBF) to promote the formation of calcium phosphate crystals, what would be a proof of bioactivity of the nanofibers.

The article is written in concise manner. In my opinion the research program in this manuscript is relatively well designed and material characterization is well investigated, however, comment or explanation is required about follow questions.

1. It would be beneficial for the manuscript to show the chemical structures (formulae) of polymers used as components of nanofibers, as a Figure.

2. IR spectra (Figure 1). I think that they are unreadable. I recommend putting all spectra into one larger drawing with the characteristic bands highlighted. Those bands should be clearly indicated on the Figure and described in the text.

3. Mineralization of the fibers. In my opinion SEM micrographs are poor quality and it is hardly to see the nanosized precipitates, which are claimed by Authors. Especially micrograph F has insufficient sharpness and poor quality.

The average Ca/P ratio is about 1.33. Since such Ca/P ratio is lower than 1.67, which is typical for hydroxyapatite, it corresponds rather to octacalcium phosphate (OCP; Ca/P = 1.33). As was reported by Dorozhkin (Calcium orthophosphates in nature, biology and medicine. Materials 2009;2(2):399498 ) a similarity in crystal structure between OCP and hydroxyapatite is one of the reasons for epitaxial growth of these both phases.

4. Materials and Methods.

Methods are described in this chapter but Materials are missed.

Line 268. “Then, CIMD was dissolved with PCL in the same solution for obtaning 10 wt % solution”. This sentence is not clear. What was the concentration od PCL? Of CIMD? What was the PCL/CIMD ratio?

The sentence (line 61-65): “Several… instability and failure” is too long and incomprehensible. Please, try to divide it for two shorter sentences.

Comments on the Quality of English Language

The manuscript should be checked throughout the manuscript in order to correct grammatical and editing errors that appear in the article.

Author Response

In the manuscript Authors investigated the selected properties of the nanofibers composed of polycaprolactone and cellulose modified with imidazolium tosylate in order to incorporate cationic groups into its chains. Sodium alginate was used as a polymer bearing anionic groups to combine with nanofibers by interactions with CIMD. The product was treated by simulated body fluid (SBF) to promote the formation of calcium phosphate crystals, what would be a proof of bioactivity of the nanofibers. The article is written in concise manner. In my opinion the research program in this manuscript is relatively well designed and material characterization is well investigated, however, comment or explanation is required about follow questions.

1.  It would be beneficial for the manuscript to show the chemical structures (formulae) of polymers used as components of nanofibers, as a Figure.

Thank you for the comment. more detailed information about the chemical formulae and the functional groups involved in the interaction mechanisms have been also reported in the graphical abstract.

2. IR spectra (Figure 1). I think that they are unreadable. I recommend putting all spectra into one larger drawing with the characteristic bands highlighted. Those bands should be clearly indicated on the Figure and described in the text.

Thank you for the comment. The Figure 1 has been changed as overlay of all the spectra. Main characteristic bands have been also highlighted.

3.1 Mineralization of the fibers. In my opinion SEM micrographs are poor quality and it is hardly to see the nanosized precipitates, which are claimed by Authors. Especially micrograph F has insufficient sharpness and poor quality.

Thank you for the comment. The resolution of the SEM images has improved, the Figure 2F has been changed as requested. 

3.2 The average Ca/P ratio is about 1.33. Since such Ca/P ratio is lower than 1.67, which is typical for hydroxyapatite, it corresponds rather to octacalcium phosphate (OCP; Ca/P = 1.33). As was reported by Dorozhkin (Calcium orthophosphates in nature, biology and medicine. Materials 2009;2(2):399‐498 ) a similarity in crystal structure between OCP and hydroxyapatite is one of the reasons for epitaxial growth of these both phases.

Thank you for the comment. The text has been modified accordingly with your suggestion as follows “The average Ca/P ratio is in the range of octacalcium phosphate (about 1.33), which is a precursor for biological apatite formed during biomineralization”.

4.1  Materials and Methods: Methods are described in this chapter but Materials are missed.

Thank you for the comment. The materials section has been added as follows:          ”Microcrystalline cellulose, p-toluenesulfonyl chloride, sodium alginate (SA), 1-methylimidazole and Polycaprolactone (Mw 70,000–90,000) were purchrased from Sigma-Aldrich). Formic acid (85%) and acetic acid (96%) from Edwic, Egypt, 1-Butyl-3-methylimidazolium chloride from IoLiTec (Heilbronn, Germany). No purification was required for chemicals used, being of analytical quality”.

4.2 Line 268. “Then, CIMD was dissolved with PCL in the same solution for obtaning 10 wt % solution”. This sentence is not clear. What was the concentration of PCL? Of CIMD? What was the PCL/CIMD ratio?

Thank you for the comment. The sentence has been rewritten as follows: “1 g of PCL pellets was dissolved in 10 ml mixture of formic acid/acetic acid solvents (70:30 v/v) to form a clear solution (10% w/v) under continuous stirring for 120 minutes. Then, a fixed amount of CIMD was dissolved into PCL solution to obtain 80:20 (w/w) PCL/CIMD relative ratio”

4.3 The sentence (line 61-65): “Several… instability and failure” is too long and incomprehensible. Please, try to divide it for two shorter sentences.

Thank you for the comments. The sentence has been changed as follows: “Several studies suggested the use of bioactive treatments to make biologically active the surface of polymeric substrates. This is crucial to create optimal microenvironmental conditions for cells, by promoting a more efficient interfacial bonding with the surrounding tissues and limiting the uncontrolled formation of fibrous tissue at the interface, that may be relevant in terms of failure or implant instability”.

Reviewer 2 Report

Comments and Suggestions for Authors

Biomimetics-2024-2908202

Biomineralization of Polyelectrolyte Functionalized Electrospun 2 Fibers: Optimization and In vitro Validation for Bone applica-3 tions

This study investigates the potential of a novel hybrid electrospun PCL/CIMD/SA scaffold in terms of in vitro mineralization performances. The work is completed with antibacterial properties assessment.

Comments

The paper is globally difficult to read as in a draft configuration (red changes etc…). Despite this, the approach is interesting. However, some points should be completed or modified.

-why PCL ? due to degradation time issues ? what are the specifications actually ? all this should be brought at begin to better motivate the work

-why CIMFD and SA ? The advantages of these substances should be discussed in more details based on thorough literature data. What has been tested by others ?

-the Materials and Methods section is missing

-how was the scaffold produced ? how much PCL was considered vs SA and CIMFD ? what were the  electrospinning parameters (voltage, distance to collector, viscosity etc...). All this is missing 

-Fig 2: not clear what the difference is as the scale is not the same

-“….As reported in previous work, good cell adhesion and proliferation were recognized in PCL nanofibers modified with CIMD and CIMD/SA…” so what is the added value here.

-the use of SBF is standard, is that the novelty here ?

-“…with more remarkable effects in the case of PCL/CIMD/SA/CP due to the hydrophilic contribution of SA…” what is SA and its role, explain in more details ….

 -“…cell proliferation increased sig-205 nificantly after 7 days of culture…” what are the mechanisms involved, based on literature data as this has already been observed by others

-“…noteworthy is that the presence of SA promoted a significant increase in the ALP activity with respect to untreated samples or only CIMD-treated ones…” what are the mechanisms involved ?

-“…ascribable to the contribution of hydroxyl groups that can improve the interfacial interaction…” has this been proven by others ?

Author Response

This study investigates the potential of a novel hybrid electrospun PCL/CIMD/SA scaffold in terms of in vitro mineralization performances. The work is completed with antibacterial properties assessment. The paper is globally difficult to read as in a draft configuration (red changes etc…). Despite this, the approach is interesting. However, some points should be completed or modified.

Thank you, red text changes previously present into the manuscript have been removed as requested. All the new changes have been marked in yellow. 

1. why PCL? due to degradation time issues? what are the specifications actually? all this should be brought at begin to better motivate the work

PCL electrospun nanofibers were widely used in bone tissue engineering due to their optimal mechanical properties (i.e, toughness, mechanical stiffness) that tend to be stable under physiological conditions for several months, as a consequence of their slow degradation in biological fluids [21,22]. Their combination with natural polysaccharides such as cellulose and sodium alginate allows to improve the fiber biocompatibility, contributing to the creation of a substrate highly friendly for cells and able to facilitate the transport of molecules, drugs and tissue morphogens [23]. A recent work of our group has just demonstrated that CIMD in combination with SA show interesting antibacterial properties and good in vitro response for wound healing applications [24]. In this context, it was proposed to functionalize composite fibers with a bone-like calcium phosphate layer via biomimetic treatment in order to overcome the intrinsically limited activity of these polyelectrolites to support osteogenesys [25].

2. why CIMFD and SA? The advantages of these substances should be discussed in more details based on thorough literature data. What has been tested by others?

Thank you for the question, Antibacterial properties of CIMD and SA polyelectrolytes and in vitro response for tissue engineering were reported in our previous work (Molecules 28, 21 (2023): 7305). A sentence has been included in the results and discussion section to clarify this aspect as follows:” A recent work of our group has just demonstrated that CIMD in combination with SA show interesting antibacterial properties and good in vitro response for wound healing applications [24]”.

3. Materials and Methods section is missing

Thank you for the comment. The materials section has been added as follows:          ”Microcrystalline cellulose, p-toluenesulfonyl chloride, sodium alginate (SA), 1-methylimidazole and Polycaprolactone (Mw 70,000–90,000) were purchrased from Sigma-Aldrich). Formic acid (85%) and acetic acid (96%) from Edwic, Egypt, 1-Butyl-3-methylimidazolium chloride from IoLiTec (Heilbronn, Germany). No purification was required for chemicals used, being of analytical quality”.

4. how was the scaffold produced? how much PCL was considered vs SA and CIMFD ? what were the  electrospinning parameters (voltage, distance to collector, viscosity etc...). All this is missing 

Thank you for the comment. The sub paragraph related to the preparation of fibers has been rewritten in order to include the requested information as follows: “1 g of PCL pellets was dissolved in 10 ml mixture of formic acid/acetic acid solvents (70:30 v/v) to form a clear solution (10% w/v) under continuous stirring for 120 minutes. As for blended fibers, a fixed amount of CIMD was dissolved into PCL solution to obtain 80:20 (w/w). PCL/CIMD relative ratio. PCL and PCL/CIMD solutions were processed via electrospinning by using a home-made system with vertical configuration composed of a syringe pump (KDS Series 100) connected to a metallic needle (19G) to control the solution flow rate and a power supply with single (positive) polarity (Glassman High Voltage Series - working range 0-20 kV) to control the applied voltage. Process parametes were set as follows: Voltage 17 kV, flow rate 0.5 mL/h, needle tip/ground distance of 150 mm. The All the tests were performed at room temperature (about 25 o C) and at a relative humidity of 60 ± 19 %”.

As for the SA functionalization, it was remarked that SA was used as a crosslinker to prevent fast release of CIMD due to it high water solubility. It was also observed that the samples did not gain that much weight after immersion in SA and also SBF.

5. Fig 2: not clear what the difference is as the scale is not the same

Thank you for the comment. In the Figure 2, three different typologies of samples were reported before and after SBF treatment. In the case of treated samples, images at higher magnification (scale bar 5 micron) were selected to better recognize calcium phosphate deposits onto the sample surface.   

6. “….As reported in previous work, good cell adhesion and proliferation were recognized in PCL nanofibers modified with CIMD and CIMD/SA…” so what is the added value here.

Thank you for the comment. In this work it was focused on the role of biomineralization on in vitro response. For this purpose, the study was conduited with hMSC cells in order to evaluate the biological response in osteogenic way for a potential application in bone tissue engineering. A new sentence has been included in the introduction section to highlight this aspect as follows:” A recent work of our group has just demonstrated that CIMD in combination with SA show interesting antibacterial properties and good in vitro response for wound healing applications [24]. In this context, it was proposed to functionalize composite fibers with a bone-like calcium phosphate layer via biomimetic treatment in order to overcome the intrinsically limited activity of these polyelectrolites to support osteogenesys [25]” 

7. the use of SBF is standard, is that the novelty here?

Thank you for the comment. In this work, the impact of CIDM/SA polyelectrolyte layer on promoting calcium phosphate deposition during the SBF treatment was highlighted. No studies are reported in literature about this. The results demonstrated that the CIDM/SA polyelectrolyte layer improve the surface reactivity thus promoting a more copious precipitation of calcium phosphate from SBF.

8. “…with more remarkable effects in the case of PCL/CIMD/SA/CP due to the hydrophilic contribution of SA…” what is SA and its role, explain in more details ….

Thank you for the comment. SA was used as a crosslinker to prevent fast release of CIMD due to it high water solubility. The hydrophilic properties of SA in PCL/CIMD/SA have been discussed in a previous work pf our group [Ref 24]. The citation has been now reported into the text as follows”… with more remarkable effects in the case of PCL/CIMD/SA/CP due to the hydrophilic con-tribution of SA, as reported in previous works [24]”. 

9. “…cell proliferation increased significantly after 7 days of culture…” what are the mechanisms involved, based on literature data as this has already been observed by others

Thank you for the comment, some information about the mechanisms of proliferation and differentiation have been added, as follows: “This may be attributable to an initial stage of differentiation triggered by the presence of CaP [29], that induce a metabolic shift from proliferaton to differentiation, thus limiting the proliferation [30–32]

10. “…noteworthy is that the presence of SA promoted a significant increase in the ALP activity with respect to untreated samples or only CIMD-treated ones…” what are the mechanisms involved?

Thank you for the comment, some sentences have been revised to improve the discussion about the effect of SA on cell interaction, as follows: “In this view, hMSC differentiation was evaluated after 3 and 7 days using an early mark-er, i.e., alkaline phosphatase (ALP) (Figure 6). After 3 days, an increase in ALP was recog-nized in the case of CaPs-coated samples respect to the negative controls (TCP and PCL). However, some differences can be recognized as a function of the fiber composition – e.g., SA and/or CMID, thus confirming the active role of the polyelectrolyte functionalization on influencing cell differentiation, over the presence of CaP [37, 38].”

11. “…ascribable to the contribution of hydroxyl groups that can improve the interfacial interaction…” has this been proven by others?

Thank you, for the comment, further information has been added with new references that indicates the participation of hydroxyl groups on calcium phosphates precipitation as follows: “After 7 days of culture, an increase in ALP activity was recognized in all the groups. In particular, the presence of SA concurs to promote a more remarkable increase of the ALP activity respect to only CIMD-treated ones. This is due to the contribution of hydroxyl groups able to support the formation nucleation sites for the precipitation of calcium phosphates [39-41] and improving the interfacial interaction with calcium phosphate through hydrogen bonds [42].”

Round 2

Reviewer 1 Report

Comments and Suggestions for Authors

The authors have partially taken into account my suggestions to improve the article.
I regret to say that the IR spectra still do not prove that the formation of a mineral apatite layer on the nanofibres surface has occurred. Similarly, the SEM images do not prove this.

In the IR spectra, the same bands are visible, irrespective of the material; moreover, the bands chosen by the authors as evidence for cellulose functionalisation (1361) and precipitation of apatite crystals (579) are also present in the PCL spectrum and before mineralisation.

SEM micrograph E (sample before mineralization) and F (PCL/CIMD/SA after mineralization) are very similar. It is strange.

Figure 3. Elemental mapping by EDS could have been evidence if the authors had shown mapping also for the sample before mineralisation.

Comments on the Quality of English Language

-

Author Response

The authors have partially taken into account my suggestions to improve the article. I regret to say that the IR spectra still do not prove that the formation of a mineral apatite layer on the nanofibres surface has occurred. Similarly, the SEM images do not prove this.

Thank  you for the comments. We have tried to satisfy your request by adding more comments and  further data.

In the IR spectra, the same bands are visible, irrespective of the material; moreover, the bands chosen by the authors as evidence for cellulose functionalisation (1361) and precipitation of apatite crystals (579) are also present in the PCL spectrum and before mineralisation.

Thank you for the comment. As reported from EDS analysis, the percentage of calcium and  phosphate after the mineralization is not so high. Accordingly, FTIR spectra do not clearly show the calcium phosphate peaks which are partially covered by PCL/CIMD/SA ones. In order to improve the readability of the spectra, we tried to distinguish the most promising peak at 933 cm-1 which is attributed to phosphate groups. Besides, it is important to notice that, comparative EDS results (now data samples before mineralization have been included) clearly confirm the presence of calcium phosphates onto the proposed materials,  due to the biomineralization treatment.   

SEM micrograph E (sample before mineralization) and F (PCL/CIMD/SA after mineralization) are very similar. It is strange.

Thank you for the comment. As discussed in the previous comment, it has been chosen a biomineralization treatment few invasive to limit the potential problems of inflammation due to exceeding mineral phases that can be released into the in vitro/in vivo, as reported in previous studies [Acta Biomaterialia, 9, 2, 2013, Pages 4956-4963, https://doi.org/10.1016/j.actbio.2012.09.035]. According to this point, mineral deposits cannot be easily recognized via SEM images. This has been now underlined in the results section. However, a comparative study of samples before and after treatment via EDS analyses has been performed  to validate the efficiency of proposed biomineralization process. 

Figure 3. Elemental mapping by EDS could have been evidence if the authors had shown mapping also for the sample before mineralization.

Thank you for the comment. The EDS of the control PCL/CIMD/SA was included in the manuscript.

Reviewer 2 Report

Comments and Suggestions for Authors

all my comments have been addressed. I thank the authors

Author Response

Thank you for  the positive feedback